

# Impact of Optimal Observational Time Window on Parameter Optimization and Climate Prediction: Simulation with a Simple Climate Model

A.A.Yuxin Zhao[1], B.B.Xiong Deng[1], C.C.Shuo Yang[1]

[1]College of Automation, Harbin Engineering University, Harbin, China

*Correspondence to: B.B.Xiong Deng (XiongDeng407@hrbeu.edu.cn)*

**Abstract.** Usually, an optimal time window (OTW) centred at the assimilation time to collect measured
data for an assimilation cycle, can greatly improve the CDA analysis skill. Here, with a simple coupled
model, we study the impact of optimal OTWs on the quality of parameter optimization and climate
prediction. Results show that the optimal OTWs of valid atmosphere or ocean observations exist for the
parameter being estimated and incorporating the parameter optimization will do some impact on the
optimal OTWs for the state estimation. And using the optimal OTWs can enhance the predictability
both of the atmosphere and ocean.

## 1.    Introduction

Because of the imperfect model equations, numeric schemes, and physical parameterizations, as well as
the biased model parameters, climate models always drift away from the real world (e.g., Delworth et
al., 2006; Collins et al., 2006; Zhang, 2011a,b; Zhang et al., 2012; Wu et al., 2012,2013b; Liu et al.
2014a,b; Han et al., 2014;). Parameter optimization, which includes the model parameters into control
variables, is a promising way to partly compensate for the bias of the values of the model parameters
and improve the climate predictability(e.g. Zhang, 2011a,b; Zhang et al., 2012,2013b; Wu et al.,
2012,2013; Liu et al. 2014a,b; Han et al., 2014;).

In the words of Han et al. (2013), given the importance of the balance and coherence of different model
components (or media) in coupled model initialization, it has been realized that for the purpose of
climate estimation and model initialization, data assimilation (including model state estimation and
parameter optimization) should be performed within a coupled model framework which can reasonably
simulate the interaction of major components of the earth climate system, such as the atmosphere,
ocean, land, and sea ice and give the assessment of climate changes (e.g. Chen et al., 1995; Zhang et al.,
2007; Randall et al., 2007; Chen, 2010;). And in the coupled climate system, the time scale and
characteristic variability in different media are usually different. When the observational data in one or
more media are assimilated into a model, information is exchanged among different media and between
model states and parameters of the couple system. Such an assimilation procedure can sustain the
nature of multiple time-scale interaction during climate estimation (e.g. Zhang et al., 2007; Sugiura et





al., 2008), thus producing coherent and balanced coupled model initialization and parameters that may enhance model predictability (e.g., Zhang, 2011b; Yang et al., 2013).

In each component of a coupled data assimilation system, usually an observational time window (OTW) centred at the assimilation time is used to collect measured data (valid observations) for an assimilation

cycle, assuming that all the collected data sample the observation at the assimilation time, and the assimilation scheme assimilates all of these valid observations within the OTW into the coupled model states and parameters sequentially. As the previous study (Zhao et al., 2015, manuscript submitted to J. Climate) has shown that there is an optimal OTW in each coupled component for model state estimation so that the assimilation has maximum observational information but minimum variation

inconsistency and the optimal observational time windows analyzed from the characteristic variability time scales of coupled media can significantly improve climate analysis and prediction initialization since it helps recovering some important character variability such as sub-diurnal cycle in the atmosphere and diurnal cycle in the ocean. And the larger the characteristic variability time scale is, the larger the corresponding OTW is. The model parameters are lack of direct observations and prognostic

equations, parameter optimization completely relies on the covariance between a parameter and the model state (e.g., Zhang, 2011a,b; Zhang et al., 2012; Wu et al. 2012,2013; Han et al., 2014; Liu et al. 2014a,b). Thus the observational time window (OTW) of the model state in each media of the coupled climate system will do some impact on the quality of parameter optimization and climate prediction. Questions we attempt to answer in this study are: 1) Whether or not exists an optimal OTW of

atmosphere or ocean observations for parameter optimization so that the assimilation has maximum observational information but minimum variation inconsistency? 2) What is the impact of optimal OTWs of atmosphere or ocean observations on parameter optimization and climate prediction?

In this study, with a simple coupled model and the DAEPC algorithm (Zhang et al., 2012) which is based on the ensemble adjustment Kalman filter (EAKF, e.g. Anderson, 2001; 2003; Zhang and

Anderson, 2003;), starting from the characteristic variability time scale of each coupled component and model parameter, we first identify the optimal OTW for each component and parameter optimization. Then we examine the impact of optimal OTWs on parameter optimization and climate prediction. The simple coupled model consists of chaotic (synoptic) atmosphere (Lorenz 1963) and seasonal-interannual slab upper ocean (Zhang et al., 2012) that couples with decadal deep ocean (Zhang

2011a,b). Although the simple coupled model does not have complex physics as a coupled general circulation model (CGCM), it does characterize the interaction of multiple time-scale media in the climate system (Zhang et al., 2013; Zhao et al., 2015, manuscript submitted to J. Climate). The simple model helps us understand the essence of the problem we want to address here. Using the DAEPC algorithm with the simple coupled model, we first establish a biased twin experiment framework where

the degree to which the state and parameter estimation based a certain OTW recovers the truth is an assessment of the influence of the OTW on the quality of parameter optimization and climate prediction. With this biased twin experiment framework, we identify the optimal OTW for the model parameters and examine the impact of optimal OTWs on the quality of parameter optimization and climate prediction.



This paper is organized as follow. Section 2 briefly describes the simple coupled model, the ensemble adjustment Kalman filter for state estimation and parameter optimization and the biased twin experiment framework. Then the influence of OTW on the quality of the parameter optimization and climate prediction are investigated in section 3. Summary and discussions are given in section 4.

## 2. Methodology

### 2.1 The model

Because of the complex physical processes and huge computation cost involved, it is not convenient to use a CGCM to study the influence of observational time window on the quality of parameter optimization and climate prediction (e.g., Zhang 2011a,b; Zhang et al., 2012; Han et al., 2013, 2014; Zhao et al., 2015, manuscript submitted to J. Climate). Instead, here we employ a simple decadal prediction model developed by Zhang (2011a). Same as Zhang (2011a), this simple decadal prediction model is based on the Lorenz's 3-variable chaotic model (Lorenz, 1963) and couples the three Lorenz chaotic atmosphere variables to a slab ocean model (e.g., Zhang 2011a,b; Zhang et al., 2012; Han et al., 2013,2014; Zhao et al., 2015, manuscript submitted to J. Climate) and a simple pycnocline predictive model (e.g., Gnanadesikan, 1999; Zhang 2011a,b; Han et al., 2013,2014; Zhao et al., 2015, manuscript submitted to J. Climate). This simple coupled model shares the similar fundamental features with the CGCMs to investigate the problems in this study (e.g., Zhang 2011a; Han et al., 2014; Zhao et al., 2015, manuscript submitted to J. Climate). The governing equations of this simple coupled climate model are as follow:

$$
\begin{aligned}
\dot{\mathcal{X}}_1 &= -\sigma\mathcal{X}_1 + \sigma\mathcal{X}_2 \\
\dot{\mathcal{X}}_2 &= -\mathcal{X}_1\mathcal{X}_3 + (1 + \mathcal{C}_1\omega)k\mathcal{X}_1 - \mathcal{X}_2 \\
\dot{\mathcal{X}}_3 &= \mathcal{X}_1\mathcal{X}_2 - b\mathcal{X}_3 \\
\mathcal{O}_m\dot{\omega} &= \mathcal{C}_2\mathcal{X}_2 + \mathcal{C}_3\eta + \mathcal{C}_4\omega\eta - \mathcal{O}_d\omega + \mathcal{S}_m + \mathcal{S}_s\cos(2\pi t/\mathcal{S}_{pd}) \\
\Gamma\eta &= \mathcal{C}_5\omega + \dot{\mathcal{C}}_6\omega\eta - \mathcal{O}_d\eta
\end{aligned}
\tag{1}
$$

where the five model variables represent the atmosphere $(\mathcal{X}_1, \mathcal{X}_2$ and $\mathcal{X}_3)$ and the upper ocean ($\omega$ for the upper slab ocean) and the deep ocean ($\eta$ for the deep ocean pycnocline). A dot above the variable denotes the time tendency. The atmosphere model variables are of high frequency and the standard values of their relevant parameters $(\sigma, k$ and $b)$ set as 9.95, 28 and 8/3, respectively, which can sustain the chaotic nature of the atmosphere in reality. The slab ocean model state is a lower frequency variable. And the parameters $\mathcal{O}_m$ and $\mathcal{O}_d$ in the equation of $\omega$ represent the heat capacity and damping coefficient of the upper ocean, respectively. The frequency of $\omega$ is much lower than that of the atmosphere model variables, thus the slab ocean model state must have a much slower time scale than atmosphere model variables and the heat capacity should be much larger than the damping rate, namely $\mathcal{O}_m \gg \mathcal{O}_d$. Here the parameters $(\mathcal{O}_m, \mathcal{O}_d)$ set as (10,1), which represent that the time scale of the slab ocean is defined as ~O(10), 10 times of the atmospheric time scale ~O(1). While the $\mathcal{S}_m + \mathcal{S}_s\cos(2\pi t/\mathcal{S}_{pd})$ represents the external forcing, $\mathcal{S}_{pd}$ is set as 10, which represents that the period of the external forcing is similar



with the time scale of the upper ocean and defines the time scale of the model seasonal cycle. $\mathcal{S}_m$ and $\mathcal{S}_s$ define the magnitudes of the annual mean and seasonal cycle of the external forcing, which are not sensitive to the coupled model and set as (10,1). The coefficients $\mathcal{C}_1$ and $\mathcal{C}_2$ in the equations of $\mathcal{X}_2$ and ω are chosen as (0.1,1), which realize the coupling between the fast atmosphere and the slow slab

ocean, and the $\mathcal{C}_1$ represents the slab ocean forcing on the atmosphere and $\mathcal{C}_2$ in contrast. In addition, $\mathcal{C}_3$ and $\mathcal{C}_4$ denote the linear forcing of the deep ocean and the nonlinear interaction of the slab and deep ocean. For guaranteeing the dominant role of the interaction between atmosphere and the slab ocean in the slab ocean model, the magnitudes of $\mathcal{C}_3$ and $\mathcal{C}_4$ are smaller than that of $\mathcal{C}_2$ and set as 0.01 in this study. In this simple coupled climate model, the seasonal cycle is defined as 10TUs, and thus a model

year (decade) is defined as 10(100)TUs. In the words of Zhang (2011a), the deep ocean pycnocline model state η represent the anomaly of the deep ocean pycnocline depth and its time tendency equation is derived from the two-term balance model of the zonal-time mean pycnocline (Gnanadesikan, 1999). And in the equation of η, the parameter Γ is a constant of proportionality and the ratio of Γ and $\mathcal{O}_d$ defines the time scale of η. Because η is a deep ocean variable, its time scale is larger than that of the

slab ocean variable ω. Here the time scale of η is defined as ~O(100), 10 times of the time scale of ω, namely Γ is set as 100. $\mathcal{C}_5$ and $\mathcal{C}_6$ denote the linear forcing of the slab ocean and the nonlinear interaction of the slab and deep ocean. Also for guaranteeing that the linear interaction is stronger than the nonlinear interaction and the nonlinear interaction in the slab ocean model is stronger than that in the deep ocean pycnocline model, the number magnitudes of $\mathcal{C}_5$ is larger than that of $\mathcal{C}_6$ and the

magnitudes of $\mathcal{C}_4$ is larger than that of $\mathcal{C}_6$. Here, $\mathcal{C}_5$ and $\mathcal{C}_6$ are set as (1,0.001). So in this study, the standard values of the parameters including in this simple coupled model $(\sigma, k, b, \mathcal{C}_1, \mathcal{C}_2, \mathcal{O}_d, \mathcal{O}_m, \mathcal{S}_m, \mathcal{S}_s, \mathcal{S}_{pd}, \Gamma, \mathcal{C}_3, \mathcal{C}_4, \mathcal{C}_5, \mathcal{C}_6)$ are set as (9.95,28,8/3,0.1,1,1,10,10,1,10,100,0.01,0.01,1,0.001)(e.g., Zhang 2011a,b; Zhang et al., 2012; Han et al., 2013,2014; Zhao et al., 2015, manuscript submitted to J. Climate).

In this paper, the simple coupled model uses the fourth-order Runger-Kutta (RK4) time-differencing scheme (e.g., Han et al., 2014; Zhao et al., 2015, manuscript submitted to J. Climate), which can be described as following Eq.(2). Where $k_0 - k_3$ represent four time levels. φ represents state variables in Eq.(1). $\Delta t$ is the time interval (Here $\Delta t = 0.01$TU) and $\mathcal{F}$ is the right term of state variables in Eq.(1).

$$
\begin{aligned}
k_0 &= \Delta t \mathcal{F}(\varphi^n) \\
k_1 &= \Delta t \mathcal{F}(\varphi^n + k_0/2) \\
k_2 &= \Delta t \mathcal{F}(\varphi^n + k_1/2) \\
k_3 &= \Delta t \mathcal{F}(\varphi^n + k_2) \\
\varphi^{n+1} &= \varphi^n + \frac{1}{6}(k_0 + 2k_1 + 2k_2 + k_3)
\end{aligned}
\tag{2}
$$

Zhang (2011b) illustrated that, this simple coupled climate model with the standard parameters described above can share the common feature that different components of various timescales interact with each other to develop climate signals with the real world climate system. In the words of Han et al. (2014), in this simple coupled model, the transient atmosphere attractor, the slow slab ocean and the even-slower deep ocean interact to produce synoptic decadal timescale signals (see Zhang, 2011a; Han

et al., 2014; Zhao et al., 2015, manuscript submitted to J. Climate).

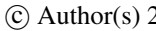



### 2.2 Ensemble coupled data assimilation for state and parameter estimation

In the words of Zhang (2011a), an ensemble filter uses the error statistics evaluated from ensemble model integrations, such as the error covariance between model states to extract observational information to adjust the model states for state estimation (e.g., Evensen, 1994, 2007; Anderson, 2001; Hamill et al., 2001; Zhang, 2011a,b; Zhang et al., 2012; Wu et al., 2012,2013; Han et al., 2014; Liu et a., 2014a,b;). And the ensemble-evaluated covariance between the model states and model parameters can also be used to estimate the model parameters (e.g., Anderson, 2001; Annan and Hargreaves, 2004; Annan et al., 2005; Askoy et al., 2006a,b; Evensen, 2007; Hansen and Penland, 2007; Kondrashov et al., 2008; Tong and Xue, 2008b; Yang and Delsole, 2009; Delsole and Yang, 2010; Zhang, 2010; Zhang 2011a,b; Wu et al., 2012,2013; Han et al., 2014; Liu at al., 2014a,b; Zhang et al., 2015;). In this study, the authors employ the DAEPC algorithm (Zhang et al., 2012), which is based on the standard ensemble adjustment Kalman filter (EAKF, e.g., Anderson 2001; 2003; Zhang and Anderson, 2003; Zhang et al., 2007), to implement the coupled state estimation and adaptive parameter optimization. And the EAKF algorithm is a sequential implementation of ensemble Kalman filter (Kalman, 1960; Kalman and Bucy, 1961) under an "adjustment" idea. The assumption of independence of observation error allows the EAKF to sequentially assimilate observations into corresponding model states and parameters (Zhang, 2003; Zhang et al., 2007). On one hand the EAKF algorithm can provide much computation convenience for data assimilation and parameter optimization, on the other hand it can maintains much the non-linearity of background flows as much as possible (e.g., Anderson, 2001; 2003; Zhang and Anderson, 2003).

Based on the two-steps of EAKF (Anderson, 2001; 2003), the first step computes the observational increment (e.g., Zhang et al., 2007; Wu et al., 2012,2013) using

$$\Delta \mathcal{Y}_{k,i} = \left( \bar{y}_k^u + \Delta \hat{y}_{k,i} \right) - \mathcal{Y}_{k,i}^{\mathcal{P}} \qquad (3)$$

where $\Delta \mathcal{Y}_{k,i}$ denotes the observational increment of the $i$th ensemble member of the $k$th observation $\mathcal{Y}_{k,i}$; $\bar{y}_k^u$ is the posterior mean of the $k$th observation; $\Delta \hat{y}_{k,i}$ is updated ensemble spread of the $k$th observation for the ensemble member; $\mathcal{Y}_{k,i}^{\mathcal{P}}$ is the $i$th prior ensemble member of the $k$th observation. Once the observation increment is computed as above, it can be projected onto related model variables and parameters using the following uniform linear regression formula:

$$\Delta Z_{k,i} = \frac{\mathcal{C}(Z^{\mathcal{P}}, \mathcal{Y}_k)}{\sigma_k^2} \Delta \mathcal{Y}_{ki,} \qquad (4)$$

Where $\Delta \mathcal{Y}_{ki,}$ represents the observation increment of $\mathcal{Y}_{ki,}$ and $\mathcal{C}(Z^{\mathcal{P}}, \mathcal{Y}_k)$ defines the error covariance between the prior ensemble of the model state or parameter variables and the model estimated observation ensemble. $\sigma_k$ is the standard deviation of the model estimated ensemble of $\mathcal{Y}_k$. The term $\Delta Z_{k,i}$ is the contribution of the $k$th observation to the model state or parameter variables $Z$ for the $i$th ensemble member (e.g., Zhang 2011a.b; Zhang et al., 2012; Wu et al., 2012,2013;  Han et al., 2014; Liu et al., 2014a,b;). The application of Eq.(4) to the coupled model states when the reliable observations are available implements CDA for state estimation in a straight forward manner (Zhang et al., 2007; Zhang 2011a;). However because of the model parameters are lack of the internal variability and prognostic equation, effective parameter estimation is very difficult before the uncertainty of model states have been sufficiently constrained by observation. And in order to achieve a signal-dominant



parameter-state covariance and have an enhancive parameter correction with observation information, the application of Eq.(4) for parameter optimization must be delayed until the coupled model state estimation reaches a quasi-equilibrium, where the errors of model states become mainly contributed from model parameter errors. (e.g., Zhang, 2011a,b; Zhang et al., 2012; Wu et al., 2012,2013; Han et al., 2014; Liu et al., 2014a,b;). Once the model parameters are optimized by the Eq.(4), the updated parameters will further promote the state estimates in the next data assimilation cycle.

In addition, the inflation scheme is essential for the parameter optimization. In this study, the inflation scheme for the DAEPC algorithm follows Zhang et al. (2012), which is formulated as

$$\tilde{\beta}_\ell = \bar{\beta}_\ell + \max\left(1, \frac{\alpha_0 \sigma_{\ell,0}}{\sigma_\ell \sigma_{\ell,t}}\right)\left(\beta_\ell - \bar{\beta}_\ell\right) \tag{5}$$

Same as Zhang et al. (2012), $\beta_\ell$ and $\tilde{\beta}_\ell$ represent the prior and the inflated ensemble of the $\ell$th parameter. $\sigma_{\ell,t}$ and $\sigma_{\ell,0}$ are the prior spreads of $\beta_\ell$ at time $t$ and the initial time. $\alpha_0$ is the constant tuned by a trial-and-error procedure. $\sigma_\ell$ is the sensitivity of the model state with regard to $\beta_\ell$ (see the similar experiment of model sensitivities on parameter described in Zhang (2011b) and Zhang et al. (2012)). And the $\bar{\beta}_\ell$ represents the ensemble mean. The Eq.(5) indicates that if the prior spread of $\beta_\ell$ is less than $\alpha_0/\sigma_\ell$ times the initial spread, it will be enlarged to this amount (e.g., Zhang 2011a.b; Zhang et al., 2012; Wu et al., 2012,2013; Han et al., 2014; Liu et al., 2014a,b;).

### 2.3    Biased twin experiment framework setup

In this study, a bias twin experiment framework is designed. Same as Zhang (2011a), in the biased twin experiment, while the "truth" model uses the standard parameter values listed in section 2.1, the assimilation model uses biased parameter values that have 10% overestimated error than the corresponding standard values. In this study we assume that the parameter errors are the only source of model errors. The "truth" model produces the true solution of the model states and observations are sampled from the "truth". The model starts from the initial condition (0,1,0,0,0) and integrates forward 10000TUs (1TU= 100$\Delta t$) for sufficient spin-up. After the spin-up, the model integrates for another 10000TUs for producing the "truth" and observations. The observations are produced through sampling the "true" model state values at an observational frequency and superimposed with a white noise which simulates the observational error. Here, the observational intervals of all the valid observations in this simple coupled model are all assumed to be 1 time step. (Although in the real observation system, the atmosphere observations are available more frequently than the ocean and less frequently than the time step. In this study, we are concerned about the influence of the observational time window on the quality of coupled data assimilation (including the state and parameter estimation) and the optimal observational time window for each component and model parameters. If the observational intervals are set too large, it may damage the characteristic variability. Thus, we hope the observational time interval is small as much as possible. So, for simplicity, the observational intervals are all set to be 1 time step). The standard deviations of the observation errors are 2 for $\mathcal{X}_1, \mathcal{X}_2, \mathcal{X}_3$ and 0.5 for $\omega$. And usually there is no valid observations in the deep ocean, thus no observation is available for $\eta$ (e.g., Zhang 2011a,b; Han et al., 2014; Zhao et al, 2015, manuscript submitted to J. Climate). The biased model with the biased parameters is used to produce the biased initial condition for all the following

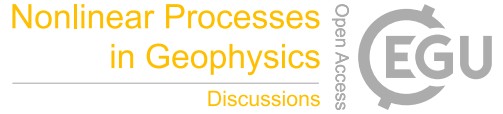

assimilation experiments. And it also starts from the initial condition (0,1,0,0,0) and spins up for 10000TUs. A Gaussian white noise with the same standard deviations as the corresponding observational errors (2 for $\mathcal{X}_1, \mathcal{X}_2$ and $\mathcal{X}_3$, 0.5 for $\omega$, 0.06 for $\eta$) is added on the model states at the end of spin-up to form the ensemble initial condition. In all of the assimilation experiments, same as the

previous studies (e.g., Zhang 2011a,b; Han et al., 2013; 2014), the assimilation intervals are set to be 5 time steps for $\mathcal{X}_1, \mathcal{X}_2, \mathcal{X}_3$ and 20 time steps for $\omega$, respectively. The total data assimilation period is 10000TUs, and parameter optimization is started after 3000TUs when state estimation reaches its "quasi-equilibrium" (e.g., Zhang, 2011a,b; 2012; Wu et al., 2012,2013; Han et al., 2014). And another 2000TUs will be the spin-up of the parameter optimization. All statistics are computed using the results

of the last 5000TUs. In this study, the observations including in the observational time windows (OTWs) will be used to sequentially adjust the model states or/and parameter being estimated at the assimilation time in all the assimilation experiments. And throughout this study, all the assimilation experiments do not use the multi-variate adjustment scheme. (As Zhao et al. (2015, manuscript submitted to J. Climate) has shown that the multi-variate adjustment scheme just using the cross

covariance will not change the characteristic variability time scales and the optimal observation time window in different components. And the multi-variate adjustment scheme using the coupling covariance cross the different media may do some impact to the characteristic variability time scales in different components, which will complex the investigation of the observational time window. Thus in this study, for simplicity, all the assimilation experiments do not use the multi-variate adjustment

scheme.)

All the assimilation experiments including in the biased experiment frameworks are all start from the ensemble initial condition created above and the model are all with the biased parameters. First, in the biased twin experiment framework, a free assimilation model control without observation constraint servers as a reference for the evaluation of any assimilation with an observation constraint within the

biased experimental framework, called the model control (CTL). The coupled data assimilation experiment only assimilates the observations into model states without parameter optimization and the observational time windows, called state estimation only (SEO). And the SEO experiment using the optimal observational time windows (both the atmosphere and slab ocean, called ATM-OTW and OCN-OTW, respectively) in different media, is called SEO_With_OOTW. The experiments assimilate

the observations into model states and one single model parameter, called Single Parameter Estimation (SPE). As the SPE experiment without using the observational time windows, is called SPE_Without_OTW. Among the SPE experiments, the model states estimation and parameter optimization use the same observational time windows, called SPE_With_S_P_OTW. (In the SPE_With_S_P_OTW experiment, the states estimation and parameter optimization use the same

observational time windows. So there are two observation time windows for state and parameter estimation, namely the atmosphere State-Parameter observational time window (ATM-S-P-OTW) and Slab Ocean State-Parameter observational time window (OCN-S-P-OTW)). As the SPE experiment, only the state estimation or the parameter optimization uses the observational time windows, called SPE_With_S_OTW and SPE_With_P_OTW, respectively. (In the SPE_With_S_OTW experiment,

there are two observational time windows for state estimation, namely ATM-S-OTW and OCN-S-




OTW. And In the SPE_With_P_OTW experiment, there are two observational time windows for parameter optimization, namely ATM-P-OTW and OCN-P-OTW.)

In order to investigate the impact of the OTWs on climate prediction, we conduct some forecast experiments aiming to the five cases (SPE_Without_OTW, SPE_With_S_P_OTW,

SPE_With_S_OTW, SPE_With_P_OTW, SEO_OOTW). Table 1 lists the details of the twin experiment frameworks.

Same as Zhang and Anderson (2003), based on the trade-off between computation cost and assimilation quality, the ensemble size of 20 is chosen in this study (e.g., Zhang and Anderson, 2003; Zhang, 2011a,b; Zhang et al., 2012; Wu et al., 2012,2013; Zhang et al., 2015; Han et al., 2013; 2014;

Zhao et al, 2015, manuscript submitted to J. Climate).

### 3.    Impact of OTWs on the quality of the parameter optimization and climate prediction

In this section, under the biased twin experiment framework, we will show the influence of OTWs on the quality of the parameter optimization and climate predictability.

### 15  3.1    Characteristic variability of each component and parameter of the model

In a coupled climate model, the characteristic variability time scale at which the flow varies in different media is different. Sustaining the characteristic variability of the different components in the coupled model is the key to improving the quality of the coupled data assimilation. If an OTW is too large, it may damage the characteristic variability of the model, which will adversely impact the state

estimation. Thus, the OTW must be smaller than the corresponding characteristic variability time scale of the component. As the previous study (Zhao et al., 2015, manuscript submitted to J. Climate) has shown that the characteristic variability time scale of the model atmosphere ($\mathcal{X}_2$), upper ocean ($\omega$) and deep ocean ($\eta$) is about 1TU, 10TUs (1 model year) and 100TUs (1 model decade), respectively, through the power spectrum analysis. And the optimal Atmosphere observation time window (ATM-

OTW) and slab ocean observation time window (OCN-OTW) in the CDA_NoMul_OTW_bias experiment are about 3 and 21, respectively, (Here 3/21  represent that there are 3/21 valid observations beside each side of the Central Time Point (right at the assimilation time, namely there are 7/43 valid observations within the ATM-OTW/OCN-ATW) which is much smaller than the corresponding characteristic variability time scale of each component (100/1000). And the larger the characteristic

variability time scale is, the larger the corresponding optimal OTW is.

In this study, each model parameter in the coupled climate model takes a globally  uniform value, which do not change with time. Thus, we can think that the characteristic variability time scales of the model parameters in this study are all about 0TU. And the characteristic variability time scales of the model states are much larger than those of the model parameters. Using the observational information

of the model states to correct the biased model parameters will do some impact on the optimal observational time windows for the parameter optimization and climate prediction.



### 3.2 Optimal observational time windows for the model parameters

In order to study the impact of the observational time window on the quality of the parameter optimization, we should set two observational time windows for the parameter optimization. Here there are two ways to establish these two observational time windows: the model state and parameter estimation use the same or different observational time windows (ATM-OTW and OCN-OTW). When the model state and parameter estimation use the same observation time windows, there are two observational time windows for model state and parameter optimization (ATM-S-P-OTW and OCN-S-P-OTW). Otherwise, there are two observational time windows for the state estimation and another two observational time windows for the parameter optimization (ATM-S-OTW, OCN-S-OTW and ATM-P-OTW, OCN-P-OTW). But the four OTWs case not only complex the adjusting and computation process, but also it may complex the investigation in this study and be not suitable in reality. Thus in this study we only consider the case that the state estimation and parameter optimization use the same OTWs.

In the biased experiment framework, the coupled models set with the biased values of all the parameters and initialized from the perturbed ensemble initial conditions. The CTL experiment is set without the observational constraint and the model ensemble is integrated for 10000TUs, serving as the reference to other assimilation experiments in the biased twin experiment framework. The SEO experiment just assimilate the observations at the assimilation time into the model states without observational time windows and parameter optimization and the SEO_OOTW experiment uses the optimal observational time windows (3 and 21 for ATM-OTW and OCN-OTW, Zhao et al., 2015, manuscript submitted to J. Climate) for state estimation. And in this study we consider that characteristic variability time scales of all the model parameters are same (0 TU). So for simplicity, we can choose one parameter to investigate the impact of the observational time windows on the parameter optimization. In this study we choose the model parameter $k$ (the standard value is 28 and overestimated value is 2.8, namely the RMSE of the parameter is 2.8 if without parameter optimization) to conduct the SPE experiments. As the SEO experiment does, the SPE_Without_OTW just assimilate the observations at the assimilation time into the model state and parameter estimation. In the SPE_With_S_P_OTW, the state estimation and parameter optimization use the same observational time windows. In the SPE_With_S_P_OTW experiment, there are two OTWs, which collect the valid atmosphere and slab ocean observations, called the ATM-S-P-OTW and OCN-S-P-OTW, respectively. The first step we set the OCN-S-P-OTW as 0 (means a single observation and no window). Next, we use L to represent the length of an OTW, meaning that the OTW includes L valid observations at either side of the assimilation time, so the total number of observations within the OTW is 2L+1. Fig. 1 shows the root-mean-square errors (RMSEs) of $\mathcal{X}_{1,2,3}$ $\omega$ and $\eta$ ,where the $\mathcal{X}_{1,2,3}$ is the arithmetical average of the RMSEs of the atmosphere model sates.

From Fig.1, we can learn that the optimal ATM-S-P-OTW and OCN-S-P-OTW for state estimation are about 2 and 10, which represent that the atmosphere (slab ocean) OTW includes 5 (21) valid atmosphere (slab ocean) observations and the lengthen of the optimal ATM-S-P-OTW (OCN-S-P-OTW) is 4(20) time steps, with the evidence of the lowest RMSEs of the $\mathcal{X}_{1,2,3}$ and $\omega$. And the RMSEs



of the $\mathcal{X}_{1,2,3}$, ω and η are respectively reduced about 30%(50%), 62%(21%) and 13%(2%) compared to the experiment of SPE_Without_OTW (SEO_With_OOTW). The optimal OTWs for state estimation are smaller than the corresponding ones in the SEO_With_OOTW experiment (3 and 21, respectively). But for the parameter being estimated, the optimal OTWs are 0 and 20, respectively. And the RMSE of the parameter being estimated ($k$) is reduced about 37% but increase about 28% compared to the experiment of SEO and SPE_Without_OTW, respectively, when using the optimal OTWs for state estimation (2 and 10).

We conduct another two experiments which only use the OTWs for state estimation or parameter optimization, namely the SPE_With_S_OTW and SPE_With_P_OTW experiment. And in the SPE_With_S_OTW experiment, we only use the OTWs for state estimation and assimilate the observations at the assimilation time into the parameter being estimated. And the results are shown as Fig. 2.

Figure 2 shows that the optimal OTWs (ATM-S-OTW and OCN-S-OTW) for state estimation are about 1 and 17, respectively, with the evidence of the lowest RMSEs of the $\mathcal{X}_{1,2,3}$ and ω. And the RMSEs of the $\mathcal{X}_{1,2,3}$, ω and η are respectively reduced about 20% (42%), 61% (17%) and 15%(2%) compared to the experiment of SPE_Without_OTW (SEO_With_OOTW). Also the optimal OTWs for state estimation are smaller than the corresponding ones in the SEO_With_OOTW experiment (3 and 21, respectively). But the optimal OTWs for the parameter being estimated are about 0 and 6, respectively. And the RMSE of the parameter being estimated ($k$) are reduces about 38.4% but increases about 11% compared to the experiment of SEO and SPE_Without_OTW, respectively, when using the optimal OTWs for state estimation (1 and 17).

In the SPE-P-OTW experiment, we only use the OTWs for parameter estimation and assimilate the observations at the assimilation time into the model states. The results are as Fig. 3.

Also from Fig. 3, we can learn that the optimal OTWs (ATM-P-OTW and OCN-P-OTW) for state estimation are 0. The results are as same as the SPE_Without_OTW experiment. But for the RMSE of the parameter being estimated, the optimal OTWs are about 0 and 20, respectively. And the RMSE of the parameter are reduced less than 2% compared to the experiment of SPE_Without_OTW experiment when the OTWs are set as 0 and 20, respectively.

The results of above experiments show that the optimal OTWs (ATM-S-P-OTW and OCN-S-P-OTW in the SPE_With_S_P_OTW; ATM-S-OTW and OCN-S-OTW in the SPE_With_S_OTW) for state estimation are smaller than the corresponding ones in the SPE_OOTW experiment. And the RMSE of the model states are reduced greatly when using these optimal OTWs for state estimation. But these optimal OTWs are not optimal for the parameter optimization. And the optimal OTWs (ATM-S-P-OTW and OCN-S-P-OTW in the SPE_With_S_P_OTW; ATM-P-OTW and OCN-P-OTW in the SPE_With_P_OTW) for parameter optimization are about 0 and 20, respectively.

The ATM-S-OTW and OCN-S-OTW aims to projecting more of the observational information of state variables onto the model state estimation and then do some impact on the parameter estimation with the observations at the assimilation time. And the ATM-P-OTW and OCN-P-OTW aim to projecting more of the observational information of state variables onto the model parameter being estimated and then do some impact on the model states in the next assimilation cycle. So adjusting the ATM-S-OTW and



OCN-S-OTW (ATM-P-OTW and OCN-P-OTW) will do some impact on the estimation of the model parameter (states).

Each parameter in the coupled climate model in this study takes a globally uniform value and the characteristic variability time scales can be considered as 0, which are much smaller than those of the model states and cause that the optimal OTWs for state estimation are smaller than the corresponding ones in the SEO_With_OOTW experiment (3/21). And the optimal OTWs of atmosphere observations for parameter optimization are much smaller than those of slab ocean observations, which is owing to characteristic variability time scales of the atmosphere model states are much smaller than that of the slab ocean model state. And when using the optimal OTWs of slab ocean observations for parameter optimization, the RMSE of the parameter being estimated are reduced slightly (less than 5%), which is owing to that the parameter ($k$) is not sensitive to the slab ocean observations.

### 3.3    Impact of the OTWs on climate prediction

Compared to the SEO_Without_OOTW experiment, above three experiments improve the quality of state or parameter estimation to some degree, but we are not sure that which case (SEO_OOTW, SPE_Without_OTW, SPE_With_S_P_OTW (OTWs are set as 2 and 10, respectively), SPE_With_S_OTW (OTWs are set as 1 and 17, respectively), SPE_With_P_OTW (OTWs are set as 0 and 20, respectively); the RMSEs of the model states in the SPE_With_S_P_OTW case is smallest and the RMSE of the model parameter being estimated in the SPE_With_P_OTW case is smallest) is of the best skill of prediction. Thus we will conduct some prediction experiments to investigate the impact of the OTWs on the climate prediction.

We launch 20 forecasts (each forward up to 50TUs (5000 time steps)) with the initial conditions selected every 50TUs apart during 8000-9000TUs. And in this twin experiment framework, we evaluate forecast skills using the anomaly correlation coefficient (ACC) and root-mean-square-error (RMSE) of forecasts verified with the "truth" (Zhang 2011b; Zhang et al. 2012). And the ACCs and RMS errors of typical "weather" forecasts ($\mathcal{X}_2$, in 1.5TUs, for instance), SI ($\omega$, in 5-10TUs) and decadal ($\eta$, in 50-100TUS) prediction are shown in Fig. 4。

With the improved initial conditions, the SPE_Without_OTW, SPE_With_S_OTW, SPE_With_P_OTW and SPE_With_S_P_OTW case greatly enhance the predictability of both atmosphere and ocean, evidenced with much higher ACC and lower RMS error compared to the SEO_OOTW case. If an ad hoc value of 0.6 ACC is use to characterize the time scale of a valid forecast/prediction (e.g., Hollingsworth et al., 1980; Zhang 2011a; Zhang et al. 2012;), the SPE_Without_OTW, SPE_With_S_OTW, SPE_With_P_OTW and SPE_With_S_P_OTW case extend a valid forecast of the atmosphere by once (beyond 0.6 TU from 0.3TU). Also the valid predictions for the upper ocean are extended by 4%, 8%, 4% and 12%, respectively, for the above four cases. The valid predictions for the deep ocean are extended by about 15%. And the predictability of the SPE_With_S_OTW and SPE_With_S_P_OTW case are slightly better than that of the SPE_With_P_OTW and SPE_Without_OTW case. The SPE_With_S_OTW and SPE_With_S_P_OTW case provide more accurate initial conditions but slight less accurate parameters than the SPE_With_P_OTW and SPE_Without_OTW case, which suggest that the initial conditions





play an more important role than the model parameter for the climate prediction when the accuracy of the parameter being estimated is lower (in all the SPE experiments, only the parameter ($k$) is estimated and other parameters are all biased).

Above results show that the optimal observation time windows for state estimation or/and parameter optimization can enhance the predictability both of the atmosphere and ocean. And the reason why the improvement of the predictability of the ocean is not obvious is that in all the SPE experiments only one parameter has been estimated and the parameter ($k$) is not sensitive to the ocean model states.

### 4.    Summary and discussion

The errors in the values of parameters in a coupled climate model are a source of model bias that causes the model to drift away from the real world. As the previous study (Zhao et al., 2015, manuscript submitted to J. Climate), in each component of a coupled data assimilation system, an observational time window centred at the assimilation time is used to collect measured data for an assimilation cycle, assuming that all the collected data sample the observational information that is

assimilated into the coupled model. The optimal observational time window for each component exists so that CDA can recover the most accurate climate signals with more observational information but minimum inconsistence of time variations in each coupled component. And he use of optimal observational time windows analyzed from the characteristic variability time scales of coupled media can significantly improve climate analysis and prediction initialization since it helps recovering some

important character variability such as sub-diurnal cycle in the atmosphere and diurnal cycle in the ocean.  Thus in this study the impact of the observational time window on the parameter optimization and climate prediction has been thoroughly examined using the DAEPC algorithm which is based on the ensemble adjustment Kalman filter consisting of a simple coupled model. The optimal observational time windows of valid atmosphere or ocean observations exist for the parameter being

estimated. And when the state estimation and parameter optimization use the same OTWs (which is suitable in practical applications), the optimal OTWs are smaller than the corresponding ones of the case only state estimation using the OTWs and without parameter optimization, which is owing to that each parameter of this coupled model takes a globally uniform value and its characteristic variability time scale is smaller than those of the coupled model components.  And the larger the characteristic

variability time scale of the model states that the observations samples from is, the larger the corresponding OTW for parameter optimization is. The optimal observational time windows for state estimation or/and parameter optimization can enhance the predictability both of the atmosphere and ocean. The simple model results suggest that when a general coupled circulation model (CGCM) is combined with the climate observing system, the use of optimal observational time windows can

significantly improve climate analysis and prediction initialization.

Although the optimal observational time window for state and parameter estimation has shown great improvement in this simple coupled model, serious challenges still exist when it is applied to CGCMs to improve the accuracy of the state and parameter estimations and the skill of climate prediction. First, the characteristic variability time scale in different components of the CGCM is impacted by many



other unknown factors owing to the complex physics. Characteristic variability in a CGCM needs to be thoroughly analyzed before an optimal observational time window is determined. Second, in this study we assume that all the valid observations including in the observational time window are equal weight to make contribution to coupled model state and parameter estimation. In fact, the further the

observation is away from the assimilation time, the less contribution it makes to the state and parameter estimation. So how to measure the weights of the observations including in the observational time windows in the complex CGCM remains to be resolved. In addition, the multi-variate adjustment scheme with the coupling covariance between the model states in different media is the important effective methods to improve the accuracy of coupled model state and parameter estimation. So the

impact of the optimal observational time window on the multi-variate adjustment scheme using the coupling error covariance should be investigated in the future studies.

### Acknowledgements

Thanks go to Drs. Shaoqing Zhang at GFDL for his comments and suggestions at the early version of

this manuscript. The authors would like to thank Drs. Chang Liu, Xuefeng Zhang, Xinrong Wu, Wei Li, Lianxin Zhang for their generous helps. Conversation with Drs. Xue Du, Renfeng Jia, Wang Li, Di Wu and Ting Zhao led to modifications of many for the first version of the manuscript. This work was supported by the National Natural Science Foundation of China (No.51379049), the Fundamental Research Funds for the Central Universities of China (No.HEUCFX41302, No.HEUCFD1505), the

Young College Academic Backbone of Heilongjiang Province (No.1254G018), and the Scientific Research Foundation for the Returned Overseas Chinese Scholars, Heilongjiang Province (No.LC2013C21).

### APPENDIX

Following the previous studies (e.g., Collins 2002; Zhang et al. 2013a), the root-mean-square error (RMSE) and anomaly correlation coefficient (ACC) base on a set of forecast experiments are calculated as following:

$$\text{ACC}(\tau) = \frac{1}{N} \frac{\sum_{j=1}^{N}\left[\hat{X}'_{f,j}(\tau) - \overline{\hat{X}'_f}(\tau)\right]\left[\hat{X}'_{t,j}(\tau) - \overline{\hat{X}'_t}(\tau)\right]}{\delta_f(\tau)\delta_t(\tau)} \tag{A1}$$

$$\text{RMSE}(\tau) = \sqrt{\frac{1}{N}\sum_{j=1}^{N}\left[\left[\hat{X}'_{f,j}(\tau) - \hat{X}'_{t,j}(\tau)\right]\right]^2} \tag{A2}$$

Where the superscript prime represents an anomaly value of variable $X$ at the lead time $\tau$ of the truth (denoted by the subscript $t$) and forecast (denoted by the subscript $f$). N represents the total number of the forecast experiments for each forecast case (in this study N is set as 20). The overbar and $\delta$ represent the average and the standard deviation of the anomaly values, respectively (Collins 2002; Zhang et al., 2013a, Han et al., 2013).






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



Table 1: List of all the experiments used in this study

| Abbreviation | Description | Observation time windows |
|---|---|---|
| CTL | Bias model without observation constrain | Not using the observational time windows |
| SEO | Bias model with model states estimation only and without parameter optimization and using the observational time window | Not using the observational time windows |
| SEO_With_OOTW | Bias model with model states estimation only and without parameter optimization and using the optimal observational time window | ATM-OTW<br><br>OCN-OTW |
| SPE_Without_OTW | Bias model with state estimation and single parameter optimization and without using the observational time window | Not using the observational time windows |
| SPE_With_S_P_OTW | Bias model with state estimation and single parameter optimization and with using the observation time window, the state estimation and the parameter optimization using the same observation time windows | ATM-S-P-OTW<br><br>OCN-S-P-OTW<br><br>State estimation and parameter optimization use the same observational time windows |
| SPE_With_S_OTW | Bias model with state estimation and single parameter optimization and with using the observational time window, only the state estimation using the observational time windows | ATM-S-OTW<br><br>OCN-S-OTW<br><br>Only state estimation using the observation time window |

| SPE_With_P_OTW | Bias model with state estimation and single parameter optimization and with using the observation time window, only the parameter optimization using the observational time windows | ATM-P-OTW<br><br>OCN-P-OTW<br><br>Only parameter optimization using the observational time windows |
|---|---|---|
| $F$ (SEO_With_OOTW) | Forecast experiment for the SEO_OOTW case | ATM-OTW(3)<br><br>OCN-OTW(20) |
| $F$ (SPE_Without_OTW) | Forecast experiment for the SPE_Without_OTW case | Not using the observational time windows |
| $F$ (SPE_With_S_P_OTW) | Forecast experiment for the SPE_With_S_P_OTW case | ATM-S-P-OTW (2)<br><br>OCN-S-P-OTW(10)<br><br>State estimation and parameter optimization use the same observational time windows |
| $F$ (SPE_With_S_OTW) | Forecast experiment for the SPE_With_S_OTW case | ATM-S-OTW(1)<br><br>OCN-S-OTW(17)<br><br>Only state estimation using the observational time windows |
| $F$ (SPE_With_P_OTW) | Forecast experiment for the SPE_With_P_OTW case | ATM-P-OTW(0)<br><br>OCN-P-OTW(20)<br><br>Only parameter optimization using the observational time window |

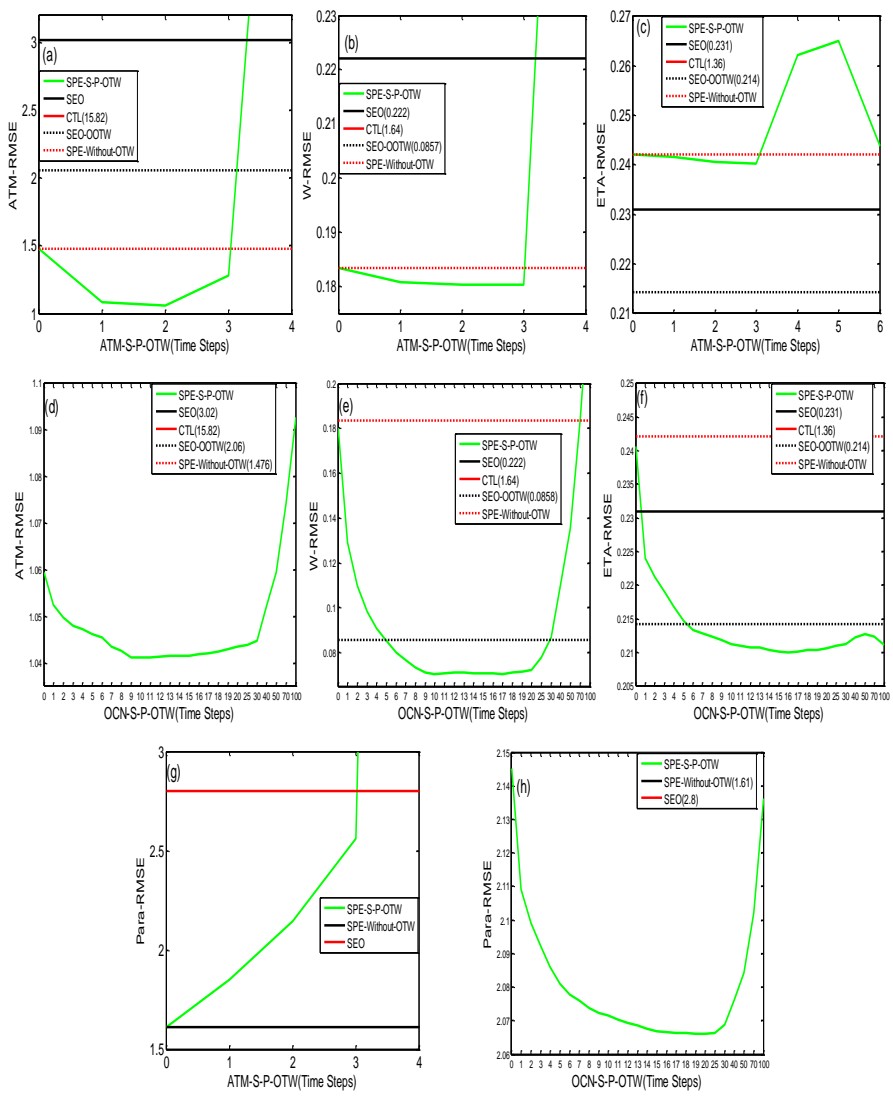

Figure 1. Variations of RMSEs of the *a)* atmosphere states $x_{1,2,3}$ (namely the average RMSE of the atmosphere states $x_1$, $x_2$ and $x_3$), *b)* the upper-ocean variable $w$, *c)* the deep ocean psycnocline depth anomaly $\eta$ and *g)* the parameter ($k$) with respect to the lengthen of the ATM-S-P-OTW, respectively, at the condition that the OCN-S-P-OTW is set as 0. And *defh)* represent the RMSEs of the $x_{1,2,3}$, $w$, $\eta$ and parameter $k$ with respect to the lengthen of the OCN-S-P-OTW, respectively, with the optimal





ATM-S-P-OTW as 2. Here the RMSEs of CTL experiment are 15.82 for $x_{1,2,3}$, 1.64 for $w$ and 1.36 for $\eta$.

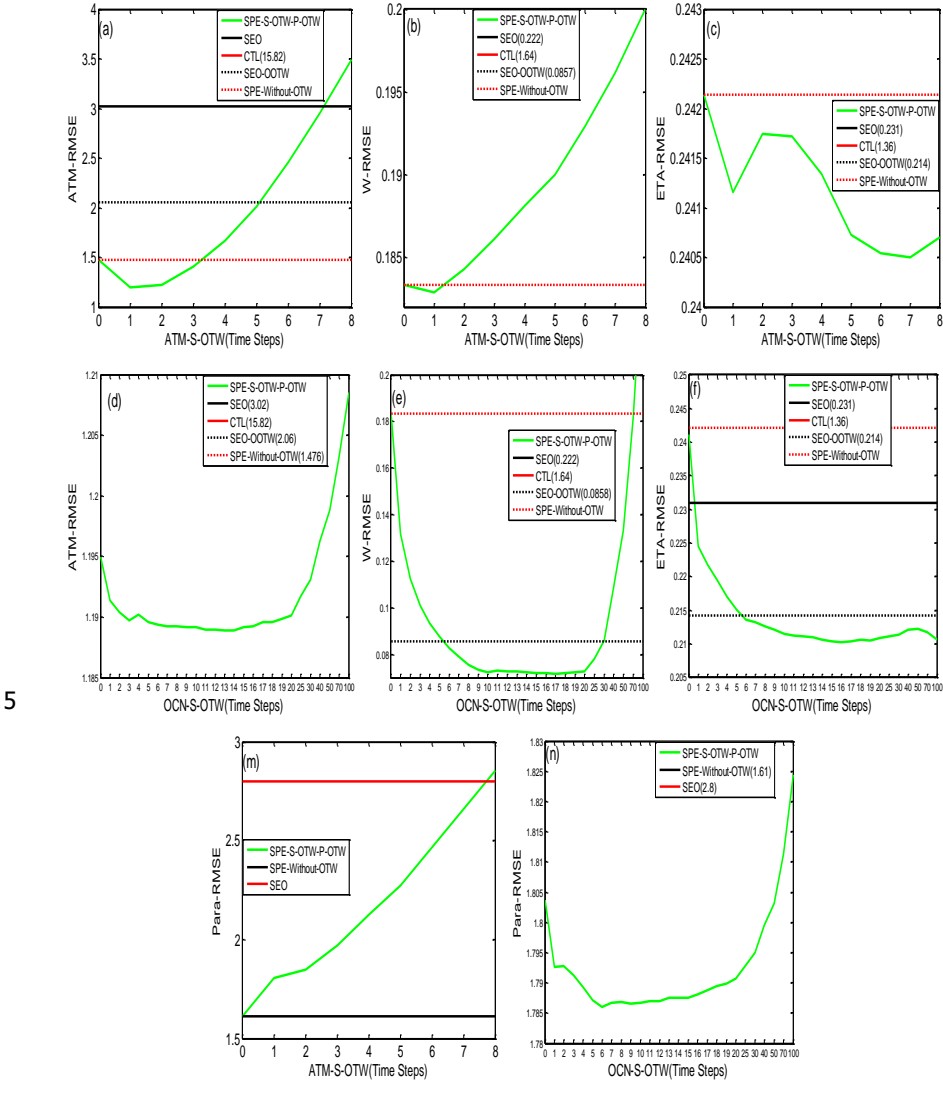

Figure 2. The same as Fig. 1 but only using the observational time windows for state estimation. *abcg)* represent the RMSEs of the $x_{1,2,3}$, $w$, $\eta$ and parameter $k$ with respect to the lengthen of the ATM-S-OTW, respectively, at the condition that the


OCN-S-OTW is set as 0. And *defh)* represent the RMSEs of the $x_{1,2,3}$, $w$, $\eta$ and

parameter $k$ with respect to the lengthen of the OCN-S-OTW, respectively, with the

optimal ATM-S-OTW as 1.

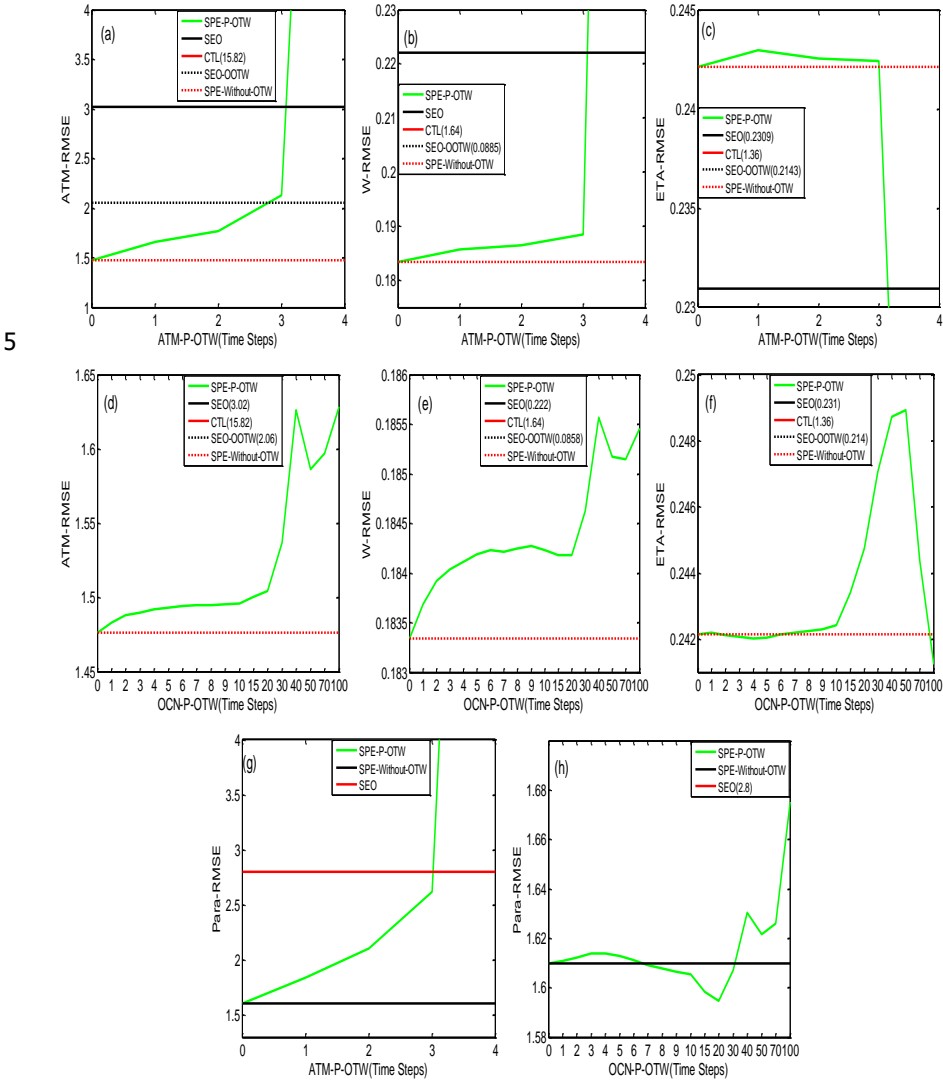

Figure 3. The same as Fig.1 but only using the observational time windows for
parameter estimation. *abcg)* represent the RMSEs of the $x_{1,2,3}$, $w$, $\eta$ and parameter $k$
with respect to the lengthen of the ATM-P-OTW, respectively, at the condition that





the OCN-P-OTW is set as 0. And *defh)* represent the RMSEs of the $x_{1,2,3}$, $w$, $\eta$ and parameter $k$ with respect to the lengthen of the OCN-P-OTW, respectively, with the optimal ATM-P-OTW as 0.

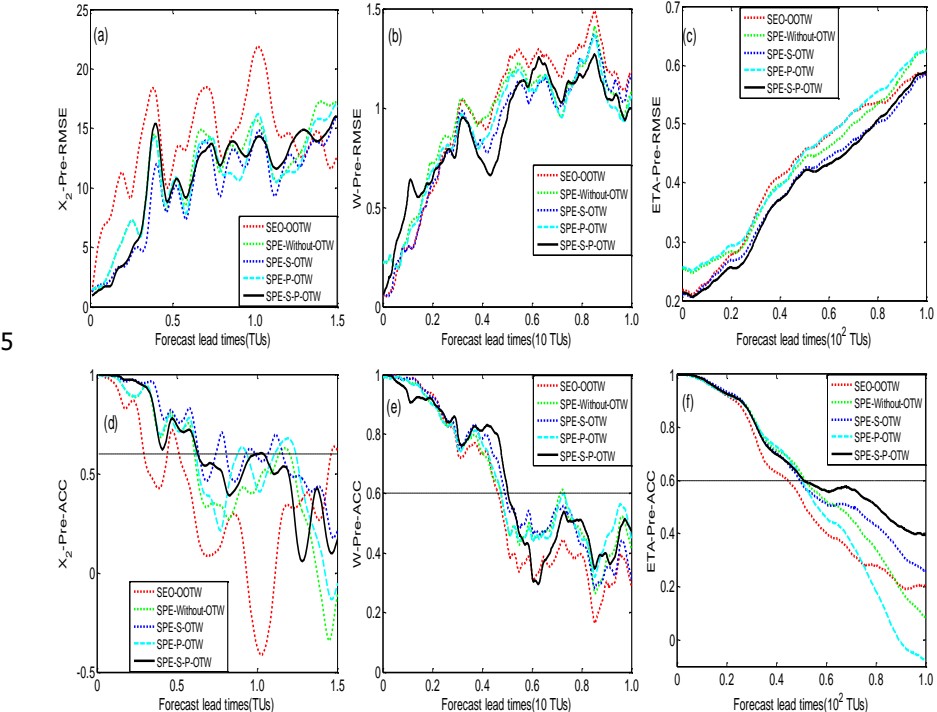

Figure 4. Variation of *abc)* the RMSEs and *def)* anomaly correlation coefficients (ACCs) of the forecasted ensemble mean of $x_2$, $w$ and $\eta$, respectively, with the forecast lead time based on 20 forecast cases initialized from the initial condition and model parameters produced by the SPE_With_S_P_OTW (black-solid line), SPE_With_S_OTW ( blue-dotted line) , SPE_With_P_OTW (cyan-blue-dashed line), SEO_With_OOTW (red-dotted line) and SPE_Without_OTW (green-dotted line) case. And the thin dotted black lines mark a 0.6 ACC level in the *def)* panels.