# Peer review of "Impact of Optimal Observational Time Window on Parameter Optimization and Climate Prediction: Simulation with a Simple Climate Model"

_Nonlinear Processes in Geophysics, 2015_

## Referee Comment (RC1) · Anonymous Referee #1 · 3 Feb 2016

This study concludes that windowing data assimilation improves forecasts, when the models/dynamics in question have multiple scales. The main focus of the paper is on the role played by model parameters and their choices in achieving more fidelity when compared to a known "truth" time series. They also explore errors in uncertainties associated in the initial conditions.

To make their case the create a evolutionary multiscale dynamics problem with some basis in a couple ocean/atmosphere flow.

Does the paper contain new and significant results?

No. Windowing is a well-known strategy in time series analysis and in dynamic forecasting. It is seldom used because unless you know what the answer should be, it can lead to very fragile outcomes. In parameter and in data assimilation in general a less fragile approach is to use locality and tapering in the corvariance, as well as regularization as a preconditioner.

The persistent allusion to optimality is troubling, when there is no theoretical basis for this conclusion, and empiricism is unacceptable evidence when the experiment considers a particular model. Further, optimality is pressumed to be measured in terms of an L2 metric. Perhaps at long time scales this is adequate, but this is the easy part of the problem; there is no way to assess this since the model itself has not been thoroughly analyzed theoretically.

There is no doubt that they have demonstrated that optimal windowing can play a role in changing the outcomes of an assimilation process, particularly if the problem is multiscale. This is shown on a single equation in a very narrow range of operating parameters. To demonstrate this on an equation like the Navier Stokes equation, for example, justifies consideration of a single equation. However, the model in question does not raise to the level of this sort. So one would need to demonstrate results more generally. The implication that the results generalize is unacceptable in explicit or implied form. This needs to be carefully demonstrated.

I have to conclude that this paper is unacceptable in its present form for publication.

Is the paper of an international standard?

I am not sure that the issue of whether this paper conforms to international vs national standards apply, and which nationality is being implied.

If what is being asked is whether this paper conforms to scientific standards, unfortunately, the answer is no. The reasons are given above.

Is the presentation clear and concise?

The authors could have compressed significantly the details of the model and the background. Instead they should have spent more real estate at clarifying their tests and to a significantly deeper analysis of the results.

Does the paper put the obtained results into context, with relevant references? Is the length of the paper appropriate? See above

Is the text fluent and precise?

No. In addition to a plethora of acronyms and unnecessarily baroque symbols, the paper also needs to be revised to improve the grammar.

Are the title and the abstract pertinent and understandable to a wide audience?

Yes, however misleading: Optimal does not mean finding a window in an ad-hoc way that leads to preferred results on a specific model.

Are all figures necessary and of appropriate quality?

They are fine, but the captions could have been improved by directing the reader to what they should conclude from these, reinforcing the text.

---

## Referee Comment (RC2) · Anonymous Referee #2 · 23 Feb 2016

Dear Editor, I have read the manuscript "Impact of Optimal Observational Time Window on Parameter Optimization and Climate Prediction: Simulation with a Simple Climate Model" by Zhao et al.

This paper deals with the optimal choice of the time window in order to better estimate the parameters of a toy climate model. The model is a low-dimensional system characterized by different time scales.

Reading the manuscript it appears that the ratio between the largest and the smallest time scales is O(100).

The authors test a data assimilation procedure based on the Ensemble Kalman filter

in order to estimate the model parameters through noisy observations. The technique of formally transform the model parameters as state (constant in time) variable is well-known and it is surely the best approach to the problem analyzed by the authors.

The original part of the work regards to the optimal choice of the time window used to collect different observations for the assimilation step. I must confess that I found the original part of the paper deeply unclear. Very often the English is incorrect (e.g. complex used as a verb) and prevent from understanding the procedure adopted by the authors.

Here is a list of the main issues:

a) it is not clear if the different observations collected during the OTW are assimilated as they were all contemporary. This is in my opinion incorrect. There are several tricks to assimilate together non-contemporary observations but it is not clear to me if the authors apply one of them.

b) The authors present several setup using different acronyms. The text is very hard to follow and probably the authors should focus on lower number of cases. Moreover some statements are really difficult to follow. For example in Sec. 3.2 Pag. 9 the line from 1 to 13 the authors speak about four OTW but the details are completely unclear to me.

c) The results shown in the figures suggest the existence of a best value for OTW but also in this case the results are presented in a confused way.

After the authors strongly modify the manuscript increasing the English quality and the clarity of their findings and procedure the paper may be considered again for publication.

---

## Author Comment (AC1) · 28 Apr 2016

Dear reviewerïijŇ Thank you very much for your generous comments and help, which are very important to improve the quality of this paper and future studies and investigations! This study "Impact of Optimal Observational Time Window on Parameter Optimization and Climate Prediction: Simulation with a Simple Climate Model" is a subsequent investigation of the previous paper "Impact of Optimal Observational Time Window on Coupled Data Assimilation: Simulation with a Simple Climate Model", which has been submitted to the Journal of Climate. And this paper aims to investigate the impact of the observational time windows (OTWs) on the quality of the parameter optimization and climate prediction. You know that the observational time window is not a

new concept. But I should tell you our thoughts about the OTWs in this study. Normally if we want to estimate the model states at 8a.m. using the EnkF, we will just assimilate the observations right at 8a.m. Other observations beside the assimilation time (For example the observations at 7:55 a.m. or 8:05 a.m.) will be ignored, which is a serious waste of the observational information. So in this study we create the observational time windows that center at the assimilation time (8 a.m.) and collect the observations at both sides of the center time point (right at the assimilation time). And we assume that all the observations including in the OTW are all sampled at the assimilation time (8 a.m.) and assimilate all of them into the model states and parameter being estimated sequentially. But we do not know how to decide the optimal length of OTW, which can mostly improve the quality of the parameter optimization and climate prediction. To investigate the impact of the OTWs on the quality of the parameter optimization and climate prediction, we using a simple coupled climate model without complex dynamics and huge computational cost. And in this study, we do not want to say that the optimal OTWs for climate and parameter estimation are accurate numbers (different models have various results). We just want to show the relationship between the optimal OTW and the corresponding characteristic variability time scale. And the results are generic, not specious for this particular simple coupled model. To investigate the essence of this problem and avoid complexing this study, we just use this simple coupled model with many simplification, which I think will help us more easily get the common conclusion and provide a guideline when the real observations are assimilated into a coupled general circulation model for improving climate analysis and prediction initialization. I do not whether this paper is of an international standard, but I just want to show that the investigation's results and conclusions may provide some guideline for CGCMs. In this paper, we just show the common conclusion from the experiments. As to the complex reason and deeply analysis, they will be deeply investigated and given in the future study. Now the revised paper has greatly improved the grammar and words. And I am so sorry that the previous ones are not fluent and misleading to you and others. As to the title, the "optimal" does not means an accurate number for a particular model.

Different models have various OTWs for different coupled components. We just want to say how to get the "optimal" and its impact and common conclusion on the coupled data assimilation, parameter optimization and climate prediction.

We just want to show some simple results and common conclusion in our experiments and hope that our explanation can answer your questions and comments. And attach please found our revised paper. Thank you very much for your generous suggestions to this manuscript again!

Sincerely yours, Xiong Deng and Co-authors
* * *